# The Unfolded Protein Response Sensor IRE1 Regulates Activation of In Vitro Differentiated Type 1 Conventional DCs with Viral Stimuli

**DOI:** 10.3390/ijms241210205

**Published:** 2023-06-16

**Authors:** Bernardita Medel, José Ignacio Bernales, Alonso Lira, Dominique Fernández, Takao Iwawaki, Pablo Vargas, Fabiola Osorio

**Affiliations:** 1Laboratory of Immunology and Cellular Stress, Immunology Program, Institute of Biomedical Sciences, Faculty of Medicine, University of Chile, Santiago 8380453, Chile; bernardita.medel@ug.uchile.cl (B.M.); jose.bernales@ug.uchile.cl (J.I.B.); alonso.lira@ug.uchile.cl (A.L.); dominique.fernandez.q@gmail.com (D.F.); 2Division of Cell Medicine, Department of Life Science, Medical Research Institute, Kanazawa Medical University, 1-1 Daigaku, Uchinada, Kahoku 920-0293, Ishikawa, Japan; iwawaki@kanazawa-med.ac.jp; 3Leukomotion Lab, Université Paris Cité, INSERM UMR-S1151, CNRS UMR-S8253, Institut Necker Enfants Malades, F-75015 Paris, France

**Keywords:** dendritic cells, cDC1s, unfolded protein response, IRE1, proinflammatory cytokines

## Abstract

Type 1 conventional dendritic cells (cDC1s) are leukocytes competent to coordinate antiviral immunity, and thus, the intracellular mechanisms controlling cDC1 function are a matter of intense research. The unfolded protein response (UPR) sensor IRE1 and its associated transcription factor XBP1s control relevant functional aspects in cDC1s including antigen cross-presentation and survival. However, most studies connecting IRE1 and cDC1 function are undertaken in vivo. Thus, the aim of this work is to elucidate whether IRE1 RNase activity can also be modeled in cDC1s differentiated in vitro and reveal the functional consequences of such activation in cells stimulated with viral components. Our data show that cultures of optimally differentiated cDC1s recapitulate several features of IRE1 activation noticed in in vivo counterparts and identify the viral analog Poly(I:C) as a potent UPR inducer in the lineage. In vitro differentiated cDC1s display constitutive IRE1 RNase activity and hyperactivate IRE1 RNase upon genetic deletion of XBP1s, which regulates production of the proinflammatory cytokines IL-12p40, TNF-α and IL-6, *Ifna* and *Ifnb* upon Poly(I:C) stimulation. Our results show that a strict regulation of the IRE1/XBP1s axis regulates cDC1 activation to viral agonists, expanding the scope of this UPR branch in potential DC-based therapies.

## 1. Introduction

Conventional dendritic cells (cDCs) are leukocytes responsible for coupling innate and adaptive immunity [1]. These cells are subdivided into type 1 cDC (cDC1) and type 2 cDC (cDC2) subtypes, which coordinate differential aspects of CD8^+^ and CD4^+^ T-cell immunity, respectively [2]. cDC1s have gained major attention in the field because of their heightened efficiency at cross-presenting viral and tumor antigens to CD8 T cells, thereby activating effective antiviral/antitumor cytotoxic T-cell responses [3].

One mechanism regulating cDC1 function is the axis controlled by the unfolded protein response (UPR) sensor IRE1, an ER transmembrane kinase and endoribonuclease (RNase) selectively activated by cDC1s in vivo [4,5,6]. IRE1 is activated by accumulation of misfolded proteins in the ER, where it oligomerizes and autophosphorylates, activating its RNase domain. The latter domain mediates an unconventional splicing of *Xbp1s* mRNA [7], prompting translation of the transcription factor XBP1s, a major regulator of ER biogenesis [7]. The RNase domain of IRE1 can also cleave additional mRNAs/miRNAs bearing a structured consensus sequence through a process termed RIDD (regulated IRE1-dependent mRNA decay) [8,9]. In cDC1s from lymphoid and mucosal tissues, RIDD has been shown to control expression of integrins, members of the antigen presentation machinery and survival, whereas XBP1s maintains the ER architecture [5,6]. These findings have placed the IRE1/XBP1s axis among the emerging pathways regulating cDC1 biology. However, most findings addressing the role of IRE1 in cDC1s emerge from studies in vivo, and it is unclear whether IRE1 activation also plays a role in counterparts derived from in vitro cultures. The validation of IRE1 activation using in vitro cultures would allow us to investigate, at the cellular level, the specific functions governed by this pathway in cDC1s, allowing a more exhaustive analysis of enzyme activation for potential therapeutics in DC-based therapies.

cDC1s have been historically complex to study owing to their low frequency in tissues and, until recently, the lack of an in vitro model that faithfully recapitulates in vivo features. We reported that cultures of bone marrow in presence of the cytokine FLT3L generate cDC1 equivalents that show signs of XBP1s activation [10]. However, in 2018, a study reported by Kirkling et al. demonstrated that cDC1s generated in FLT3L-containing cultures remain in an immature state and reported a novel protocol for generation of optimally differentiated cDC1s, which showed marked differences from existing methodologies in terms of maturation and functional capabilities [11,12,13]. This method consists of the coculture of bone marrow precursors with a stromal cell line expressing the Notch 2 ligand DL1 (termed ‘OP9-DL1’ cells) plus FLT3L, which yields mature cDC1s in large numbers and displays a more mature phenotype compared to the existing FLT3L cultures [11]. The advent of this methodology prompted us to investigate whether IRE1 activation is a feature that can be modeled in cDC1s fully differentiated in vitro. Along these lines, we recently reported that human cDC1s generated under this methodology spontaneously activate IRE1 RNase [14], which opens novel avenues for expanding studies of the IRE1/XBP1s axis also in mouse models.

The aim of this work is to demonstrate whether cDC1s differentiated in vitro can also be regulated by IRE1 RNase activity, as there is great interest in targeting the subset for biomedical applications [15]. We report that cDC1s generated by coculture with OP9-DL1 cells recapitulate most features of IRE1 activation noticed in in vivo counterparts. First, these cells display constitutive IRE1 RNase activity and basal expression of *Xbp1s* in steady state. In addition, as reported in tissues [5,6], genetic loss of XBP1s in OP9-DL1 cDC1s leads to hyperactivation of the RNase domain of IRE1 and induction of the RIDD branch, which regulates expression of costimulatory molecules in steady state, and the production of the proinflammatory cytokines IL-12, IL-6, and TNF and type I interferons upon activation with the viral agonist Poly(I:C). Thus, cDC1s optimally differentiated in vitro are a useful platform for the study of UPR components in the lineage, expanding potential therapeutic avenues of this signaling branch.

## 2. Results

### 2.1. OP9-DL1 cDC1s Display Comparable IRE1 RNase Activity to Spleen cDC1s

To date, evidence addressing the role of the IRE1/XBP1s axis in cDC1s has emerged mostly from studies in tissue-derived cells, and recent methodologies for generation of optimally differentiated cDC1s in vitro have opened novel avenues for mechanistic studies of the lineage. In this context, a reported protocol for cDC1 generation consisting of the coculture of bone marrow progenitors with the OP9 stromal cell line expressing the Notch ligand DL1 (OP9-DL1 cell line) [16,17] and the cytokine FLT3L (termed here ‘OP9-DL1 cDC’ culture) has been shown to promote optimal differentiation of cDC1s, closely resembling tissue counterparts [11]. Thus, we sought to investigate whether OP9-DL1 cDC1s also display the constitutive IRE1 RNase activity seen with in vivo equivalents. To this end, we used the ER stress-activated indicator (ERAI) mouse, a transgenic mouse that reports activity of the IRE1 RNase domain [18] by expressing a partial sequence of human *XBP1* (containing the splicing sites) fused to Venus fluorescent protein (VenusFP) (Figure 1A). Upon ER stress, cells from ERAI mice induce IRE1 phosphorylation and oligomerization, activating the IRE RNase domain, which in turn processes the XBP1s–VenusFP construct, leading to emission of Venus FP fluorescence [18]. The ERAI mouse line was previously validated in tissue cDCs as a tool to quantify IRE1 RNase activity in basal conditions and upon IRE1 hyperphosphorylation/RIDD activation [5,6]. We determined VenusFP fluorescence in OP9-DL1 cDC1s (gating strategy in Appendix A) and compared it with splenic cDC1s. Data in Figure 1B,C show that OP9-DL1 cDC1s displayed a high VenusFP signal, which was comparable to spleen cDC1s. Furthermore, cDC2s generated in OP9-DL1 cultures displayed low levels of VenusFP mean fluorescence intensity (MFI), also comparable to splenic cDC2s (Figure 1B). In addition, we also quantified VenusFP fluorescence in cDC1 cultures generated in the presence of FLT3L without OP9-DL1 cells (culture referred to as ‘FL-cDCs’), which was the method commonly used to generate cDC1s in cultures prior to the protocol reported by Kirkling et al. [13]. Data indicated that only OP9-DL1 cDC1s were capable of effective IRE1 RNase activation (Figure 1B,C). In addition, PCR for spliced/unspliced *Xbp1* forms showed that OP9-DL1 cDC1s expressed higher levels of *Xbp1* spliced (*Xbp1s*) than cDC2s obtained from the same culture (Figure 1D).

Interestingly, we observed that induction of IRE1 RNase activity in OP9-DL1 cDC cultures was a late differentiation event, occurring on day 8 of the culture and onwards, selectively in the cDC1 compartment (Figure 1E). These findings recapitulated the timing of IRE1 activation reported in tissue cDC1s [6]. In addition, IRE1 activation was synchronized with ER expansion in these cultures, as detected by ER-Tracker staining (Figure 1F). Altogether, these results indicated that optimally differentiated OP9-DL1 cDC1s recapitulated the extent and timing of IRE1 RNase activation noticed in in vivo counterparts and validate the system as an appropriate culture to study the role of IRE1 RNase activity in cDC1s.

### 2.2. IRE1 Activation in OP9-DL1 cDC1s Stimulated with Viral Agonists

Even though the IRE1/XBP1s axis is well-recognized to contribute to innate immune responses upon PRR engagement, most of these studies emerged from work in macrophages and DC subsets that do not belong to the cDC1 lineage [19,20,21,22,23,24,25,26,27,28]. Reports on activated cDC1s include parasite infection [29], and we reported that human/mouse cDC1s use IRE1 for optimal proinflammatory cytokine production using pharmacological IRE1 inhibitors [10,14]. To extend these findings and identify efficient IRE1 activators in differentiated cDC1s, we stimulated OP9-DL1 cDCs from ERAI mice with agonists of major families of PRRs: LPS (TLR4 agonist), Poly(I:C) (TLR3 agonist), Curdlan (Dectin-1 agonist), DMXAA (STING agonist), MDP (NOD2 agonist), and CpG (TLR9 agonist). Data in Figure 2A show that cDC1s displayed higher VenusFP fluorescence than cDC2s in both basal levels and upon activation with the microbial stimuli tested. We also observed that LPS and agonists of a viral nature (Poly(I:C), DMXAA, and CpG) further induced marked VenusFP expression in OP9-DL1 cDC1s. These agonists also induced expression of the costimulatory molecule CD86 (Figure 2B). However, this observation was not recapitulated in OP9-DL1 cDC2s, as these cells only induced VenusFP in response to CpG but showed CD86 upregulation in response to LPS, Poly(I:C), DMXAA, and CpG (Figure 2B). Thus, we concluded that cDC1s and cDC2s differed in their capacity to induce IRE1 RNase upon activation. For subsequent studies, we focused our analysis on Poly(I:C) stimulation, as it selectively activates IRE1 in the cDC1 compartment and it does not affect cell viability (Figure 2C).

Poly(I:C) is a synthetic analog of viral double-stranded RNA (dsRNA) that can activate additional UPR sensors including PERK, along with additional pathways such as the integrated stress response (ISR) in myeloid cells [24,28,30,31,32]. Therefore, we investigated whether Poly(I:C) stimulation also led to canonical UPR activation in sorted OP9-DL1 cDC1s (Figure 2D). Poly(I:C) treatment induced early expression of *Hspa5* (BiP) mRNA and the XBP1s target *Erp44* in OP9-DL1 cDC1s. In contrast, analysis of the canonical RIDD target *Bloc1s1* was unaltered upon Poly(I:C) stimulation, suggesting the IRE1-dependent mRNA decay was not induced in cDC1s by the viral agonist. Analysis of PERK/ISR targets revealed induction of *Chop* and *Gadd34* in response to Poly(I:C) stimulation. Finally, we observed late induction of the third branch of the UPR regulated by the sensor ATF6α, through its target *Herpud* in Poly(I:C)-activated OP9-DL1 cDC1s. Taken together, these results indicated that Poly(I:C) induces activation of IRE1/XBP1s and additional members of the PERK/ISR in OP9-DL1 cDC1s, in agreement with reports of other cell types [24,28,30,31,32].

### 2.3. Loss of XBP1s Regulates the Activation of OP9-DL1 cDC1s upon Stimulation with Poly(I:C)

We next determined the role of IRE1/XBP1s in OP9-DL1 cDC1s during activation with Poly(I:C). To this end, we generated OP9-DL1 cDCs from the bone marrow of *Itgax*-Cre x *Xbp1*^fl/fl^ conditional knock-out mice (referred to as ‘XBP1ΔDC mice’) [33,34]. *Itgax*-Cre mice target CD11c-expressing cells including cDC subsets [33]. OP9-DL1 cDC cultures from XBP1ΔDC mice generated normal frequencies of cDC1 and cDC2 at the end of coculture (Figure 3A). We determined the efficiency of Cre-mediated recombination in OP9-DL1 cDC1s from XBP1ΔDC mice (Figure 3B). This issue is highly relevant as previous reports indicated that cDC1s generated from FL-DC cultures do not mediate efficient Cre-dependent recombination of the IRE1 floxed allele [10]. However, data in Figure 3B shows that cultures of OP9-DL1 cDC1s generated from XBP1ΔDC bone marrow showed efficient deletion of the exon 2 of the *Xbp1* gene.

We next interrogated the role of XBP1s in the acquisition of immunogenic properties of OP9-DL1 cDC1s during activation. First, we corroborated that XBP1 loss did not impair survival upon Poly(I:C) stimulation (Appendix A). Analysis of XBP1-deficient OP9-DL1 cDC1s revealed normal expression of major histocompatibility complex I/II (MHC I/II) molecules prior and post Poly(I:C) stimulation (Figure 3C). However, expression of the costimulatory molecules CD80, CD86, and PD-L1 was reduced in XBP1s-deficient OP9-DL1 cDC1s in unstimulated conditions (Figure 3D), suggesting that the UPR branch regulates expression activation markers in cDC1s in homeostatic conditions. Upon Poly(I:C) stimulation, we observed that OP9-DL1 cDC1s from XBP1ΔDC mice expressed lower levels of CD80 expression. Next, the production of proinflammatory cytokines in response to Poly(I:C) treatment was analyzed by intracellular staining. OP9-DL1 cDC1s from XBP1ΔDC showed a significant decrease in the production of IL-12p40, TNFα, and IL-6 (Figure 3E,F). Bioactive IL-12 (termed IL-12p70) is comprised of the IL-12p35 (*Il12a*) and IL-12p40 subunits [35]. We observed that XBP1-deficient OP9-DL1 cDC1s displayed a trend toward reduction in *ll12a* transcription, suggesting a global reduction in IL-12 bioactivity (Appendix A). Finally, we analyzed expression of IFN-I, which is a key mediator eliciting effective antiviral responses [36,37,38]. Data in Figure 3G show a significant decrease in *Ifna4* and *Ifnb1* transcription in XBP1-deficient OP9-DL1 cDC1s activated with Poly(I:C). Overall, these data indicated that XBP1s coordinates the acquisition of major immunogenic features in cDC1s activated with viral stimuli.

### 2.4. Genetic Loss of XBP1s in OP9-DL1 cDC1s Results in IRE1 RNase Hyperactivation and Signs of RIDD

IRE1 RNase can exert its function through XBP1s activation or through RIDD. It has also been reported that XBP1s loss in tissue cDC1s results in RIDD counteractivation [5,6]. Thus, we interrogated whether XBP1-deficient OP9-DL1 cDC1s also show signs of RIDD activity by measuring *Xbp1*s through PCR. Although cDCs from XBP1ΔDCs mice do not produce the XBP1s on a protein level (owing to an early stop codon), these cells express the *Xbp1* mRNA sequence containing the IRE1 splicing sites, allowing assessment of IRE1 RNase activity [34]. Compared to control counterparts, XBP1-deficient OP9-DL1 cDC1s expressed higher levels of *Xbp1s* than *Xbp1u* (Figure 4A) and OP9-DL1 cDC1s from XBP1-deficient animals displayed a lower expression of the canonical RIDD target *Bloc1s1* (Figure 4B). These data suggested that XBP1-deficient OP9-DL1 cDC1s activate the RIDD branch, recapitulating observations noticed in in vivo counterparts [5,6]. Furthermore, these data opened the question as to whether cDC1 activation observed in vitro is dependent on the XBP1s or the RIDD branch.

### 2.5. Loss of the RNase Domain of IRE1 Modestly Affects OP9-DL1 cDC1 Activation

To interrogate whether OP9-DL1 cDC1 activation depends on XBP1s or RIDD, we crossed the *Itgax*-Cre with *Ern1*^fl/fl^ mice (referred to as ‘IRE1^truncDC^ mice’) [39]. These animals deleted exons 20–21 of the protein, generating a truncated IRE1 version that lack the RNase domain, thereby inhibiting XBP1s transcription and RIDD activation. We corroborated that OP9-DL1 cDC1s from IRE1^truncDC^ mice developed normally (Appendix A) and verified that these cells efficiently deleted exons 20-21 from the *Ern1* allele (Figure 5A). On a protein level, these cells generated the predicted truncated version of IRE1 (Figure 5B) [39] and lacked XBP1s expression by PCR (Figure 5C), confirming the absence of the RNase domain.

Next, we stimulated OP9-DL1 cDC1s from IRE1^truncDC^ mice with Poly(I:C) to assess the dependency of IRE1 RNase in cDC1 activation. Data in Figure 5D,E show that expression of costimulatory molecules was unaltered between OP9-DL1 cDC1s from control and IRE1^truncDC^ mice, except for CD40, which showed a significant increase in IRE1-deficient OP9-DL1 cDC1s with high doses of Poly(I:C) (Figure 5E). Furthermore, expression of IL-12p40, IL-6, and TNF upon Poly(I:C) stimulation was also not affected (Figure 5F,G) and expression of *Il12a, Ifna4*, and *Ifnb1* (Figure 5H; Appendix A) were not differentially regulated upon IRE1 RNase loss. The divergent results observed between XBP1s-deficient and IRE1 RNase-deficient cDC1s attribute a role for IRE1 RNase activation rather than XBP1s activity in the regulation of cDC1 activation, suggesting specific modes of cytokine regulation depending on the cell type (Scheme of summary findings depicted in Figure 6).

## 3. Discussion

DC activation is a complex process typified by the acquisition of an immunogenic profile competent to generate a long-lasting adaptive response [40]. Understanding the process of DC activation may open possibilities for identification of novel regulatory targets that improve the development of therapeutic strategies. In this context, the UPR has emerged as a relevant regulator of different types of DC subtypes with roles that stretch far beyond its canonical function [41]. DC development relies on XBP1s activity [4] and fully differentiated cDC1s depend on an intact IRE1/XBP1s axis to safeguard homeostasis [5,6]. cDCs in tissues also maintain active PERK signaling, indicating that this family of leukocytes keep strict control of their proteostatic programs [32]. It has also been reported that activated DCs expand their ER capacity [42] and activate the UPR in pathological settings including cancer and microbial stimulation, among others [21,27]. However, beneficial and detrimental roles have been found for the UPR in DCs, which seem to depend on the DC lineage and the inflammatory context (reviewed in [41]), and therefore, it is highly relevant to perform exhaustive studies in carefully delineated DC lineages to obtain a comprehensive understanding of the contribution of this adaptive program in DCs.

Here, we examined the role of the IRE1/XBP1s axis in cDC1s optimally differentiated from bone marrow precursors in cultures and showed that these cells recapitulate the constitutive activation of the enzyme noticed in in vivo counterparts. Furthermore, unlike FLT3L cDC1 cultures [10], OP9-DL1 cDC1s efficiently carry out Cre-mediated recombination of IRE1 and XBP1s floxed alleles, making this method a powerful tool to study genetic deletions in the regulation of cDC1 function.

We identified that Poly(I:C) induces efficient activation of the IRE1/XBP1s branch, along with the UPR and PERK/ISR components in OP9-DL1 cDC1s. It is known that the UPR is activated during viral recognition, and selective IRE1 activation is observed in various viral infection settings as a mechanism that promotes viral infection [43,44,45,46]. Our findings indicated that Poly(I:C) induced cDC1 activation, which was accompanied by an increase in IRE1 Rnase activity and expression of XBP1s targets, *Chop*, *Gadd34*, and *HerpuD*. These data were consistent with existing reports indicating that the PERK/ISR branch can also be activated in cDCs [32].

Analysis of steady state OP9-DL1 cDC1s showed that XBP1s loss decreases expression of the costimulatory molecules CD80, CD86, and PD-L1, indicating a control for the axis in the acquisition of activated features by cDC1s in steady state. Nevertheless, upon Poly(I:C) stimulation, the lack of XBP1s or IRE1 appears to be dispensable for the upregulation of these molecules, except for CD80. On the other hand, Poly(I:C) increases proinflammatory cytokine production in cultured cDC1s, and these factors show dependency on XBP1s expression, as induction of IL-12, TNF, and IL-6 and type I interferons were decreased in Poly(I:C)-stimulated XBP1s-deficient OP9-DL1 cDC1s. Remarkably, our data showed that these cytokines are produced at normal levels in cells lacking IRE1 RNase, which also lack XBP1s transcriptional activity. These data are novel, as they differ from previous observations in macrophages, which were known to depend on XBP1s transcriptional binding to the promoter regions IL-6/TNF for optimal cytokine production [26,47]. In contrast, the reduced cytokine production noticed in XBP1s-deficient OP9-DL1 cDC1s may be explained as alternative mechanisms that include IRE1 RNase hyperactivation, which we proved occurs in OP9-DL1 cDC1s from XBP1ΔDC animals. How exactly IRE1 RNase activation (induced upon XBP1s genetic loss) but not IRE1 RNase deficiency leads to changes in cytokine production is an issue that remains to be determined, and we speculate that it could be mediated directly by reported IRE1 RNA substrates, such as microRNAs [43,48], or indirectly, by compensatory cross-activation of other UPR branches such as the PERK axis [6]. Taken together, these results showed that a fine regulation of the IRE1/XBP1s axis controls cDC1 activation with viral agonists.

## 4. Materials and Methods

### 4.1. Mice

ER-stress activation indicator (ERAI) [18] and non-transgenic control (WT) littermates, XBP1^WT^ (XBP1fl/fl [34]), XBP1^ΔDC^ (XBP1fl/fl × *Itgax*-Cre [33]), IRE1^WT^ (IRE1fl/fl [39]), IRE1^truncDC^ (XBP1fl/fl × IRE1fl/fl × *Itgax*-Cre) were bred at the animal facilities of Universidad de Chile or Fundación Ciencia & Vida under specific pathogen-free conditions. Briefly, XBP1fl/fl mice allow Cre-mediated recombination of exon 2 of *Xbp1*, resulting in the absence of the transcription factor, while IRE1fl/fl mice delete exons 20–21 of the *Ern1* gene upon Cre-mediated recombination, which generates a truncated IRE1 isoform lacking the RNase domain [5,39]. All mice were kept on a C57BL/6 background. Litters with mice of both sexes at 6–14 weeks of age were used for experiments.

### 4.2. Media and Reagents

Culture media for OP9-DL1 cells consisted of MEM-alpha media supplemented with 20% FBS (Gibco, Waltham, MA, USA), Penicillin/Streptomycin 1× (Corning, New York, NY, USA), 1 mM sodium pyruvate (Gibco, Waltham, MA, USA), and 0.55 mM 2-Mercaptoethanol (Gibco, Waltham, MA, USA). Culture media for OP9-DL1 cDCs and FLT3-L cDC cultures was RPMI 1640 GlutaMAX (Gibco, Waltham, MA, USA) supplemented with 10% FBS (Hyclone), Penicillin/Streptomycin 1× (Corning, NY, USA), and 0.55 mM 2-Mercaptoethanol (Gibco, Waltham, MA, USA). FACS buffer was prepared with PBS 1X (Gibco, Waltham, MA, USA), supplemented with 1% FBS and 2 mM EDTA (Ambion, Naugatuck, CT, USA).

### 4.3. Cell Lines

OP9 cells expressing Notch ligand DL1 (OP9-DL1) were kindly provided by Dr. Juan Carlos Zuñiga-Pflucker (Sunnybrook Research Institute, Toronto, ON, Canada). OP9-DL1 cells were cultured in T75 flasks at 37 °C, 5% CO_2_ and maintained in a cell density of 10,000 cells per cm^2^. For OP9-DL1 cDC culture, OP9-DL1 cells were harvested with trypsin-EDTA, counted, and centrifuged at 400× *g*, 4 min, 4 °C. In all, 5000 cells were seeded per well in 24-well plates (alternatively, 20,000 cells were seeded in 6-well culture plates) in OP9 culture media and kept at 37 °C, 5% CO_2_ until the transfer of dendritic cell progenitors.

### 4.4. In Vitro Cultures of Bone Marrow Derived Conventional Dendritic Cells

Bone marrow precursors were collected from the femurs and tibias from C57BL/6 mice, and red blood cells were lysed prior to the culture. For the generation of “FLT3-L cDCs” [13], bone marrow cells (1 × 10^6^ cells/mL) were cultured in the presence of 150 ng/mL of FLT3-L recombinant mouse cytokine (Peprotech, Neuilly-sur-Seine, France) in 6-well culture plates in a total volume of 4 mL/well. The culture was maintained for 7–8 days at 37 °C, 5% CO_2_.

For the generation of “OP9-DL1 cDCs” [11], bone marrow cells (1 × 10^6^ cells/mL) were cultured for three days in the presence of 100 ng/mL of FLT3-L in 24-well plates. On day 3 of culture, bone marrow cells were transferred to the culture plate containing the OP9-DL1 cell monolayer. The coculture was maintained for an additional 6 to 7 days at 37 °C, 5% CO_2_ before use.

### 4.5. Flow Cytometry and Cell Sorting

Dendritic cells were stained in FACS buffer for 15 min at 4 °C in the dark using the antibodies indicated in the Appendix A. Upon staining, cells were washed using FACS buffer and stained with LIVE/DEAD^®^ Fixable Aqua viability marker (1/800 dilution in PBS, Molecular Probes, Thermofisher Scientific, Waltham, MA, USA) for 10 min at 4 °C in the dark. Finally, cells were washed with FACS buffer and resuspended in the same buffer for acquisition using a FACSVerse cytometer (BD, Franklin Lakes, NJ, USA). OP9-DL1 cDC cultures from ERAI reporter mice were analyzed on days 6, 8 and 10, and VenusFP was assessed on the FITC channel using a FACSVerse cytometer (BD, Franklin Lakes, NJ, USA). The ER membrane was stained by ER-Tracker Green (Thermofisher, Waltham, MA, USA) 100 nM staining on days 6, 8, and 10 of OP9-DL1 DC culture prior to the antibody staining, for 30 min at 37 °C in the dark. Cells were analyzed on the FITC channel using a FACSVerse cytometer (BD, Franklin Lakes, NJ, USA). For quantification of intracellular cytokines, OP9-DL1 cDCs were incubated 4 h with 1X Golgi-Plug (BD, Franklin Lakes, NJ, USA), and cells were washed and stained extracellularly as indicated above. Subsequently, cells were fixed and permeabilized for 30 min at 4 °C in the dark with the Cytofix/Cytoperm kit (BD, Franklin Lakes, NJ, USA) and then stained with intracellular antibodies in PermWash 1X (BD, Franklin Lakes, NJ, USA) for 30 min at 4 °C in the dark. Cells were then washed and resuspended in FACS buffer prior to acquisition. For cell sorting studies, dendritic cells were isolated from the spleen or from cultures and were subjected to a sorting process based on staining for cDC1 (XCR1+) or cDC2 (Sirpa+) subtypes.

### 4.6. Innate Stimuli and Dendritic Cell Activation

For measurement of VenusFP, UPR components, and costimulatory molecules in presence of innate immunity ligands, OP9-DL1 cDCs at 1 × 10^6^ cells/mL were either untreated (NT) or incubated for 16 h with innate immunity ligands at the following concentrations: LPS 100 ng/mL (Invivogen, San Diego, CA, USA), Poly(I:C) 10 μg/mL (HMW, Invivogen, San Diego, CA, USA), Curdlan 50 μg/mL (Wako, Richmond, VA, USA), DMXAA 10 μg/mL (Invivogen, San Diego, CA, USA), MDP 10 μg/mL (Invivogen), and CpG (ODN 1826) 1 μM (Invivogen, San Diego, CA, USA). For costimulatory molecule analysis, OP9-DL1 cDC1s were previously sorted using Mojosort negative selection (Biolegend, San Diego, CA, USA), following manufacture’s protocol. Briefly, OP9-DL1 cDCs were stained with Sirpα-Biotin and B220-Biotin, then incubated with Mojosort Streptavidin Nanobeads (Biolegend, San Diego, CA, USA) to negatively select cDC1s using the Mojosort Magnet (Biolegend, San Diego, CA, USA). OP9-DL1 cDC1s at 1 × 10^6^ cells/mL were either untreated (NT) or incubated for 16 h in presence of Poly(I:C) 10 or 50 μg/mL (HMW, Invivogen, San Diego, CA, USA). Cells were then harvested and used for further analysis.

### 4.7. PCR, qPCR and Primers

OP9-DL1 cDC1s at 1 × 10^6^ cells/mL were either untreated (NT) or incubated for 4 or 16 h in the presence of Poly(I:C) 10 μg/mL (HMW, Invivogen, San Diego, CA, USA). Cells were then harvested and used for further analysis. RNA was obtained using the TriPure isolation reagent (Roche, Sigma Aldrich, St. Louis, MO, USA) following the manufacturer’s instructions. Alternatively, RNA extraction was performed using the RNeasy^®^ Plus Micro kit (Qiagen, Hilden, Germany), following the protocol provided by the supplier. The RNA quality was determined by NanoDrop™Lite (Thermo Scientific™, Waltham, MA, USA) prior to use. cDNA was made using the M-MLV Reverse Transcriptase kit (Invitrogen, Waltham, MA, USA) and Brilliant II SYBR Green Master Mix (Agilent Technologies, Santa Clara, CA, USA). qPCR was analyzed in a Lightcycler^®^ 480 II (Roche). qPCR primers are indicated in the Appendix A.

For *Xbp1s* PCR, GoTaq Flexi polymerase (Promega, Madison, WI, USA) was used for amplification of Xbp1u/s using the mXbp1 unspl+spl F (5′-ACACACGCTTGGGAATGGACACAC-3′) and mXbp1 unspl+spl R (5′-CCATGGGAAGATGTTCTCTGGGG-3′) primers and actin as a loading control with the β-actin-F (5′-CTAAGGCCAACCGTGAAAAG-3′) and β-actin-R (5′-TTGCTGATCCACACATCTGCTGCTG-3′) primers. The amplicons *Xbp1u* and *Xbp1s* were separated by electrophoresis on a 1% agarose gel (Lafken, Santiago, Chile).

### 4.8. Western Blot

Proteins from 2 × 10^6^ of OP9-DL1 cDCs from IRE1^truncDC^ mice were isolated using E1A protein extraction buffer (50 mM HEPES pH 7.0 (Sigma Aldrich, St. Louis, MO, USA), 250 mM NaCl (Winkler Ltd., Santiago, Chile), 0.1% Nonidet P-40 (Sigma-Aldrich, St. Louis, MO, USA), 5 mM EDTA (Ambion^®^, Thermofisher Scientific, Waltham, MA, USA), protease inhibitor cOmplete ULTRA Tablets (Roche, Sigma-Aldrich, St. Louis, MO, USA), phosphatase inhibitor PhosSTOP (Roche, Sigma-Aldrich, St. Louis, MO, USA). Antibodies used include primary antibody for IRE1 (Rabbit, clone 14C10) (Cell Signaling, Danvers, MA, USA), β-actin (Mouse, clone 13E5) (Cell Signaling, Danvers, MA, USA), HRP-conjugated secondary antibody α-rabbit or α-mouse (Cell Signaling, Danvers, MA, USA).

### 4.9. Statistical Analysis

Cytometry data were analyzed using FlowJo software v10.4 (Treestar). For graph, average, SEM, and statistical analysis, Prism 9 (GraphPad) software was used. Differences between groups were analyzed using unpaired and paired two-tailed Student’s *t* tests and paired one-way ANOVA. Results with a P value equal to or less than 0.05 were considered significant. * *p* < 0.05; ** *p* < 0.01; *** *p* < 0.001; **** *p* < 0.0001.

## Figures and Tables

**Figure 1 ijms-24-10205-f001:**
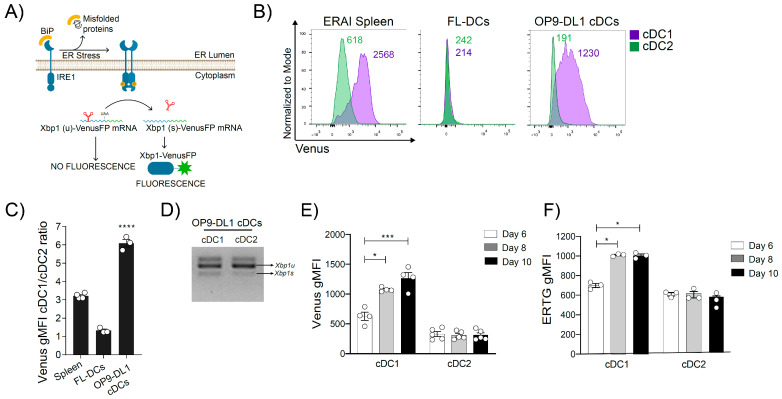
Study of IRE1 RNase activation in OP9-DL1 cDC cultures. (**A**) Representative scheme of the mouse ERAI reporter. This transgenic mouse line expresses a partial sequence of human *XBP1*, which contains the sites for IRE1-dependent splicing, fused to Venus fluorescent protein (VenusFP), thus reporting IRE1 RNase activity. (**B**) VenusFP signal of cDCs from spleen and bone marrow generated from ERAI mice. Histograms are representative of three independent experiments. The numbers in the histograms represent the gMFI of VenusFP. (**C**) Ratio of VenusFP gMFI between cDC1s/cDC2s. Each point represents an independent experiment and error bars represent mean ± SEM. **** *p* < 0.0001 (unpaired Student’s *t* test). (**D**) *Xbp1* u/s expression in isolated OP9-DL1 cDC1s and cDC2s from wild-type mice by PCR. (**E**) VenusFP expression in OP9-DL1 cDC cultures on days 6, 8 and 10. Each point represents an independent experiment and error bars represent mean ± SEM. * *p* < 0.05, *** *p* < 0.001 (paired one-way ANOVA). (**F**) Quantification of ER content by ER-Tracker green staining in OP9-DL1 cDC cultures on days 6, 8 and 10. Graphs shows the results of two independent experiments and error bars represent mean ± SEM. * *p* < 0.05 (paired one-way ANOVA).

**Figure 2 ijms-24-10205-f002:**
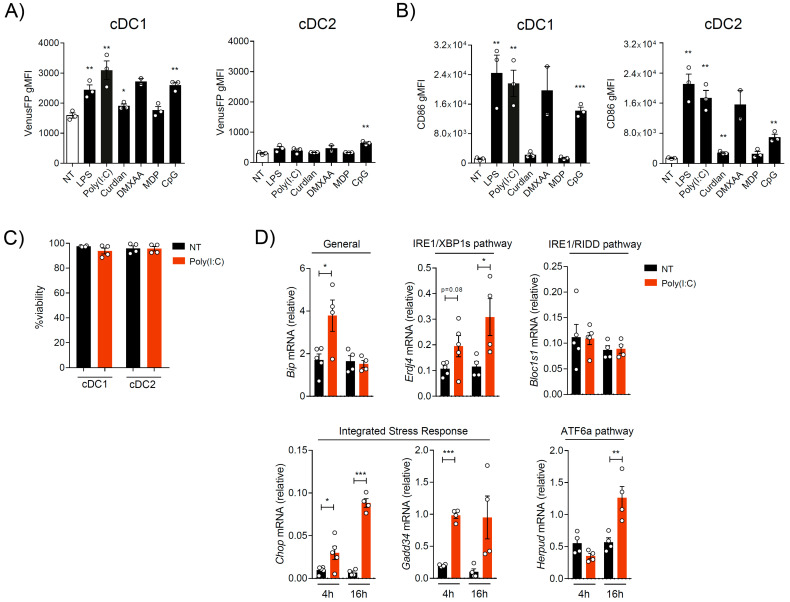
Poly(I:C) activates the RNase domain of IRE1 and additional UPR components in OP9-DL1 cDCs. (**A**) Measurement of VenusFP in OP9-DL1 cDCs from ERAI mice activated innate stimuli. Cells were untreated (NT) or treated for 16 h with LPS 100 ng/mL, Poly(I:C) 10 μg/mL, Curdlan 50 μg/mL, DMXAA 10 μg/mL, MDP 10 μg/mL and CpG 1 μM. (**B**) Cells stimulated in (**A**) were assessed for CD86 induction by flow cytometry in OP9-DL1-cDC1 (XCR1+) and OP9-DL1-cDC2 (Sirpa+) populations. (**C**) Viability of OP9-DL1 cDCs activated with Poly(I:C) (10 μg/mL) for 16 h. The percentage of live cells was analyzed by flow cytometry. (**D**) Activation of UPR members in Poly(I:C) activated OP9-DL1-cDC1s. Cells were untreated (NT) or treated with Poly(I:C) (10 μg/mL). Then, OP9-DL1 cDC1s were isolated and induction of UPR genes was quantified by qPCR at 4 h and 16 h. For all graphs, each point represents an independent experiment and error bars represent mean ± SEM. * *p* < 0.05, ** *p* < 0.01, *** *p* < 0.001 (paired Student’s *t* test).

**Figure 3 ijms-24-10205-f003:**
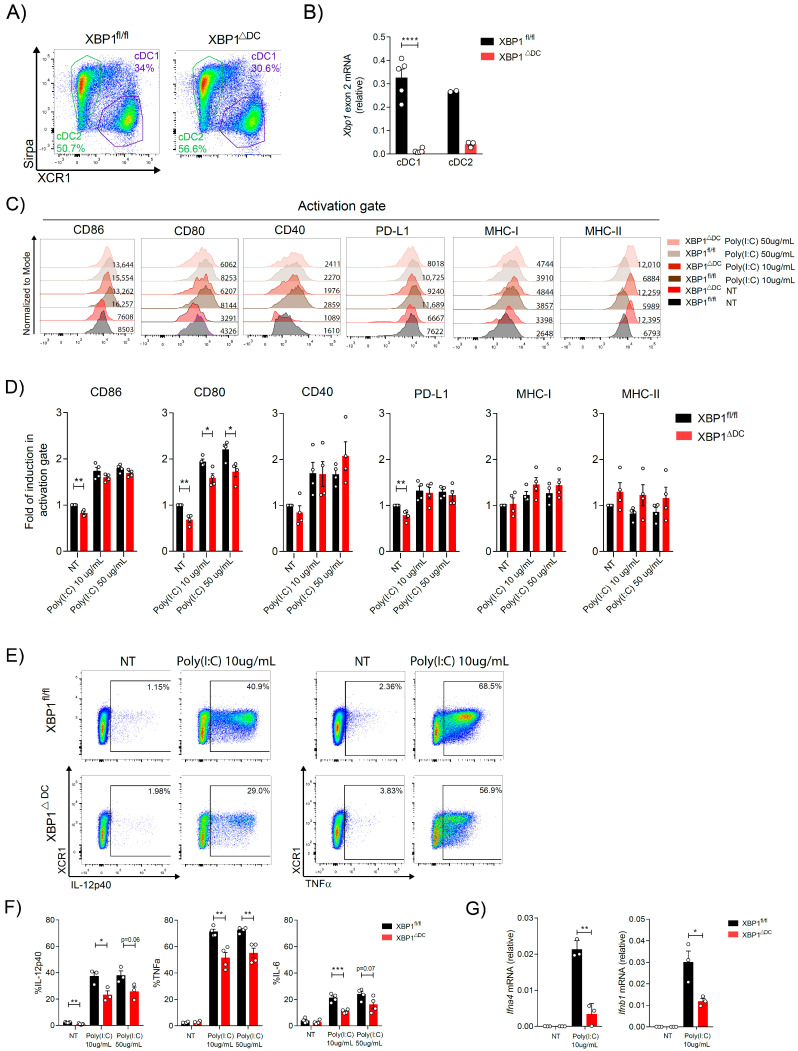
Loss of XBP1s impairs activation of OP9-DL1 cDC1s upon stimulation with viral agonists. (**A**) OP9-DL1 cDC cultures from XBP1ΔDC and control mice. (**B**) Assessment of Cre-mediated exon recombination in sorted OP9-DL1 cDCs derived from XBP1ΔDC and control mice by qPCR. (**C**) OP9-DL1 cDC1s from XBP1ΔDC and control mice were stimulated with Poly(I:C) 10 μg/mL, 50 μg/mL, or untreated (NT) for 16 h. Expression of costimulatory molecules was analyzed by flow cytometry in the MHC-II+ CD11c+ XCR1+ population (activation gate) and plotted in (**D**). Histograms are representative of three to four experiments. (**E**) Measurement of proinflammatory cytokines in Poly(I:C)-treated OP9-DL1 cDC1s from XBP1ΔDC and control mice. Cells were stimulated with Poly(I:C) 10 μg/mL, 50 μg/mL for 4 h, then intracellular cytokine production was analyzed by flow cytometry and data are plotted in (**F**). (**G**) *Ifna4* and *Ifnb1* expression was measured by qPCR in XBP1s-sufficient or -deficient isolated OP9-DL1 cDC1s stimulated with Poly(I:C) 10 μg/mL for 4 h. For all, each point represents an independent experiment and error bars represent mean ± SEM. * *p* < 0.05, ** *p* < 0.01, *** *p* < 0.001, **** *p* < 0.0001 (unpaired Student’s *t* test).

**Figure 4 ijms-24-10205-f004:**
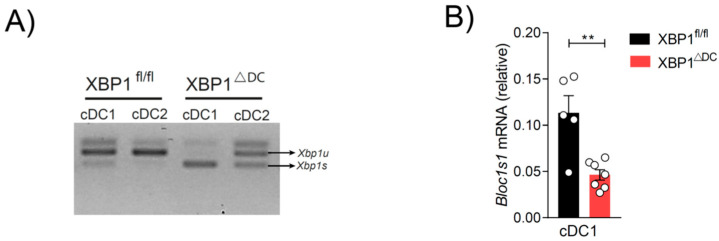
XBP1s deficiency results in IRE1 RNase hyperactivation and RIDD in OP9-DL1 cDC1s. (**A**) *Xbp1* u/s expression in isolated OP9-DL1 cDC1s and cDC2s from XBP1ΔDC and control mice by PCR. (**B**) Expression of the canonical RIDD target *Bloc1s1* in isolated OP9-DL1 cDC1s from XBP1ΔDC and control mice and quantified by qPCR. Each point represents an independent experiment and error bars represent mean ± SEM. ** *p* < 0.01 (unpaired Student’s *t* test).

**Figure 5 ijms-24-10205-f005:**
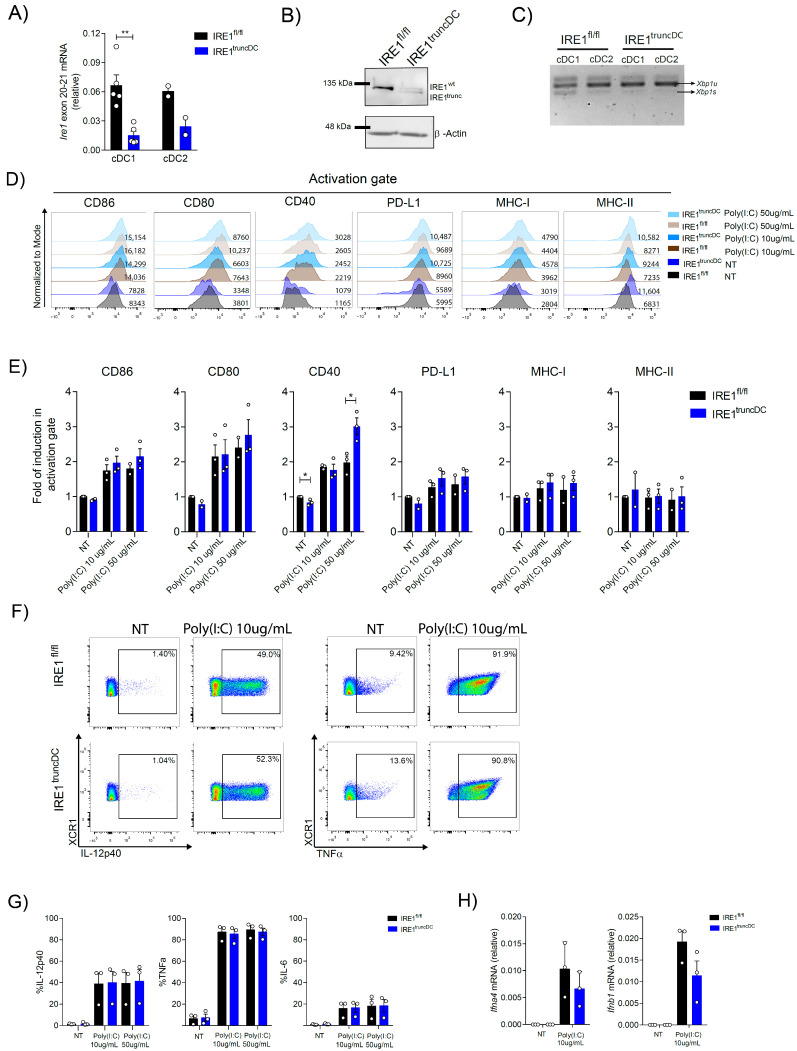
Loss of IRE1 does not influence activation of OP9-DL1 cDC1s upon stimulation with viral agonists. (**A**) Assessment of Cre-mediated exon recombination in sorted OP9-DL1 cDCs derived from IRE1^truncDC^ and control mice by qPCR. (**B**) Western blot for IRE1 expression in isolated OP9-DL1 cDC1s from IRE1^truncDC^ and control mice. (**C**) *Xbp1* u/s expression in isolated OP9-DL1 cDC1s and cDC2s from IRE1^truncDC^ and control mice by PCR. (**D**) OP9-DL1 cDC1s from IRE1^truncDC^ and control mice were isolated and stimulated with Poly(I:C) 10 μg/mL, 50 μg/mL, or untreated (NT) for 16 h. Expression of costimulatory molecules was analyzed by flow cytometry in the CD11c+ XCR1+ population and plotted in (**E**). Histograms are representative of three to four experiments. (**F**) Measurement of proinflammatory cytokines in Poly(I:C)-treated OP9-DL1 cDC1s from IRE1^truncDC^ and control mice. Cells were stimulated with Poly(I:C) 10 μg/mL, 50 μg/mL for 4 h, then intracellular cytokine production was analyzed by flow cytometry and data are plotted in (**G**). (**H**) *Ifna4* and *Ifnb1* expression was measured by qPCR in IRE1-sufficient or -deficient isolated OP9-DL1 cDC1s stimulated with Poly(I:C) 10 μg/mL for 4 h. For all, each point represents an independent experiment and error bars represent mean ± SEM. * *p* < 0.05, ** *p* < 0.01 (unpaired Student’s *t* test).

**Figure 6 ijms-24-10205-f006:**
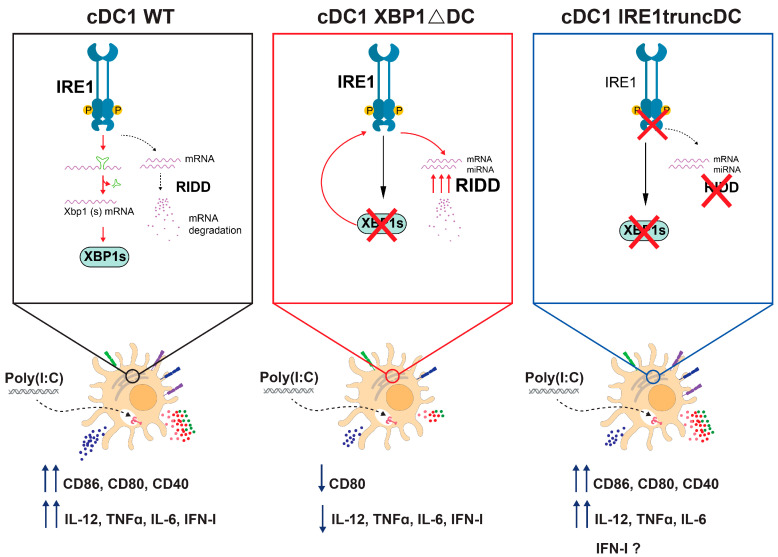
Summary of research findings. In our proposed model, Poly(I:C) recognition activates cDC1s along with promoting IRE1 RNase activation. cDC1s lacking XBP1s produce lower levels of proinflammatory cytokines and costimulatory molecules to Poly(I:C) stimulation. However, these features are not recapitulated in cells lacking IRE1 RNase, which lack both XBP1s and RIDD activity. These data suggest that an intricate regulation of the IRE1/XBP1s axis contributes to cDC1 activation with viral agonists.

## Data Availability

Data available upon request.

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
