# Peer review of "The Unfolded Protein Response Sensor IRE1 Regulates Activation of In Vitro Differentiated Type 1 Conventional DCs with Viral Stimuli"

_ijms, 2023, doi:10.3390/ijms241210205_

Round 1
Reviewer 1 Report
This paper is competently written and describes some really neat research. The XBP1s transcript with the Venus fluorescence protein is remarkable.
I only have one real criticism:
It is bad form to present graphs with same X axis beside each other with widely different Y-axis ranges. I completely understand that many of these graphs are measuring different proteins.
Further there are several graphs (in particular Fig 3 G IL-12alpha and Fig 4 H IL-12alpha where the Y-axis range is so tiny that the results are effectively zero. Okay, so IL-12 alpha is not expressed. So!
Finally, I would like to see some discussion of the role of the UPR activation in dendritic cells. Dendritic cells are key to the immune response by being antigen receivers and arm T-cells to attack invaders expressing those antigens. Why is the UPR important to differentiation of dendritic cells?
Author Response
We thank the reviewer for the input, please see attached a point-by-point reply:
This paper is competently written and describes some really neat research. The XBP1s transcript with the Venus fluorescence protein is remarkable.
I only have one real criticism:
It is bad form to present graphs with same X axis beside each other with widely different Y-axis ranges. I completely understand that many of these graphs are measuring different proteins.
R: We have now changed figures 2A, 2B, 3D, 3F, 5E and 5G according to the reviewer’ suggestion.
Further there are several graphs (in particular Fig 3 G IL-12alpha and Fig 4 H IL-12alpha where the Y-axis range is so tiny that the results are effectively zero. Okay, so IL-12 alpha is not expressed. So!
R: In our hands, the detection of the p35 subunit of IL-12 is always more difficult to detect than the p40 subunit, which is highly produced by cDC1s and illustrated in the flow cytometry analysis. However, as p35 and p40 are relevant to evaluate iL-12 bioactivity, we consider the data important. Nevertheless, we have now moved them to supplementary data (Supp. Fig. 2B and E), in line with the reviewer’ point.
Finally, I would like to see some discussion of the role of the UPR activation in dendritic cells. Dendritic cells are key to the immune response by being antigen receivers and arm T-cells to attack invaders expressing those antigens. Why is the UPR important to differentiation of dendritic cells?
R: We appreciate the input and have added the following section in the discussion:
In this context, the UPR has emerged as a relevant regulator of different types of DC subtypes with roles that stretch far beyond its canonical function (Flores Santibanez et al 2019, Cells). DC development relies on XBP1s activity (Iwakoshi et al 2007, JEM) and fully differentiated cDC1s depend on an intact IRE1/XBP1s axis to safeguard homeostasis (Osorio et al 2014, Nat. Imm; Tavernier et al 2017, Nat. Cell. Bio). cDCs in tissues also maintain active PERK signaling indicating that this family of leukocytes keep a strict control of their proteostatic programs (Mendes et al 2020, Life Sci Alliance). It is also reported that activated DCs expand their ER capacity (Everts et al 2014, Nat. Imm) and activate the UPR in pathological settings including cancer and microbial stimulation, among others (Cubillos-Ruiz et al 2015, Cell; Mogilenko et al 2018, Cell). However, beneficial and detrimental roles have been found for the UPR in DCs, which seem to depend on the DC lineage and the inflammatory context (reviewed in Flores Santibanez et al 2019, Cells) and therefore, it is highly relevant to perform exhaustive studies in carefully delineated DC lineages to obtain a comprehensive understanding of the contribution of this adaptive program in DCs.
Reviewer 2 Report
I have had the pleasure of reading "The unfolded protein ... signalling in
vitro" submitted by Medel B. et. al. which examines an in vitro method of differentiation of cDC1 and measures the effect of the unfolded protien response in the context of an anti-viral response.
Generally, the data are internally consistent and the writing is of a scholarly standard.
>Minor points -
Several times 'student's t-test' is used instead of 'Student's t-test' and 'Anova' instead of 'ANOVA'.
Section 4.2 Medium (singular) should be Media (plural).
At around Ln 325 and onwards, the font and font size are variable.
The labels on Fig. 3 F are not uniform.
Ln 22 "...IRE1 RNAse activity and noticed in in vitro counterparts"
Ln 170 "Itgax Cre mice target..." not targets
Ln 92 "We determined VenusFP fluorescence in OP9-DL1 cDC1s (gating strategy in Supp. Fig. 1A)...." There is no Supp. Fig. 1A.
Ln 195/238 Reference is made to Supp. Fig. 2. There is no Supp. Fig 2.
Check that correct gene/protein names and formats are used on figure labels.
Ln 40 "...major regulator or ER biogene..." of
The figure legend for Figure 1 makes no mention of figure element F.
The concentrations of poly I:C are quite high, up to 50ug/mL when 10ug/mL is usually used as a maximum concentration. This may explain the lack of a dose-dependent effect.
>Reporter mouse
It may improve the submission if the authors added 1-2 sentences more information on the reporter mouse or improved Figure 1A to indicate the connection between the dimerized+autophosphorylated IRE1 and the nuclease activity.
>Summary
It would improve the submission if the authors added a summary diagram of their findings.
English is generally fine, with a few typos.
Author Response
We appreciate the valuable feedback of the reviewer. Below, we provide a point-by-point reply:
I have had the pleasure of reading "The unfolded protein ... signalling in
vitro" submitted by Medel B. et. al. which examines an in vitro method of differentiation of cDC1 and measures the effect of the unfolded protien response in the context of an anti-viral response.
Generally, the data are internally consistent and the writing is of a scholarly standard.
>Minor points -
Several times 'student's t-test' is used instead of 'Student's t-test' and 'Anova' instead of 'ANOVA'.
R: We have changed the text accordingly.
Section 4.2 Medium (singular) should be Media (plural).
R: We have changed the text accordingly.
At around Ln 325 and onwards, the font and font size are variable.
R: Unfortunately, this is formatting error that was performed after submission, by the platform that changes the format to IJMS template. Our document was uploaded without template in docx and is consistent in terms of font usage.
The labels on Fig. 3 F are not uniform.
We have changed the labels of figures 3F, according to the reviewer’ suggestion
Ln 22 "...IRE1 RNAse activity and noticed in in vitro counterparts"
R: We have changed the text accordingly.
Ln 170 "Itgax Cre mice target..." not targets
R: We have changed the text accordingly.
Ln 92 "We determined VenusFP fluorescence in OP9-DL1 cDC1s (gating strategy in Supp. Fig. 1A)...." There is no Supp. Fig. 1A.
We apologize for the inconvenient, we were almost sure that we uploaded the supp figs, but in any case, they are attached to the main figures in this revised version.
Ln 195/238 Reference is made to Supp. Fig. 2. There is no Supp. Fig 2.
We apologize for the inconvenient, we were almost sure that we uploaded the supp figs at the moment of submission, but in any case, we attached them again to the main figures in this revised version.
Check that correct gene/protein names and formats are used on figure labels.
Ln 40 "...major regulator or ER biogene..." of
R: We have changed the text accordingly.
The figure legend for Figure 1 makes no mention of figure element F.
R: We have changed the text accordingly.
The concentrations of poly I:C are quite high, up to 50ug/mL when 10ug/mL is usually used as a maximum concentration. This may explain the lack of a dose-dependent effect.
R: At the beginning of the project, we did not know how XBP1 and IRE1 RNase deficient cDC1s were going to react to poly I:C stimulation, and this is why we decided to use a higher dose than 10ug/ml just in case we observed hypo-responsiveness in conditional knock-out cells. However, the reviewer is correct, we did not observe a dose effect response in wild type cDCs. In contexts of XBP1 deficiency, we observed that cDCs remain impaired at producing optimal amounts of cytokines even on the higher dose of Poly I:C.
>Reporter mouse
It may improve the submission if the authors added 1-2 sentences more information on the reporter mouse or improved Figure 1A to indicate the connection between the dimerized+autophosphorylated IRE1 and the nuclease activity.
R: We have now included the following sentences which we hope to answer the request ‘Thus, upon ER stress, cells from ERAI mice induce IRE1 phosphorylation and oligomerization activating the IRE RNase domain, which in turn process the XBP1s-VenusFP construct leading to emission of Venus FP fluorescence (Iwawaki et al 2004, Nat. Med). The ERAI mice line has been previously validated in tissue cDCs, as a tool to quantify IRE1 RNase activity in basal conditions and upon IRE1 hyperphosphorylation/RIDD activation (Osorio et al., 2014; Tavernier et al., 2017).’
>Summary
It would improve the submission if the authors added a summary diagram of their findings.
R: We have now included a graphical summary of our work (Figure 6).
Reviewer 3 Report
This study addressed the impact of IRE1 and XBP1 on poly IC (a virus agonist)-induced cDC1 activation. The rational of this study is understandable. cDC1s have gained attention due to their ability to cross-present viral and tumor antigens to CD8 T cells. Elucidation of the activation mechanism of cDC1 is critical to manipulate the cells for protection from virus infection and establishment of tumor immunity. However, this reviewer fells that the data does not support conclusion in this study.
Major points
(1) Line 301 and 302: The conclusion should be rephrased. IRE1 deficiency does not have impact on expression levels of co-stimulatory molecules and cytokine production. It does mean that this molecule is not an emerging regulator of cDC1 activation with viral agonists.
(2) The expression levels of VenusFP were significantly different between cDC1 and cDC2, whereas the expression levels of co-stimulatory molecules were similar in the two DC subsets. Therefore, it is not easy to assume that spliced XBP1 plays a crucial role in the expression of co-stimulatory molecules in cDC1. Please comment about it.
(3) Please explain the novelty of XBP1-related data.
Others
(1) Title should be rephrased.
(2) Fig. 3C: Please include FACS histogram data of CD80, CD40, PD-L1.
(3) Line 205: Is it Fig.4?
(4) The legend title of Fig. 4: Please check whether this title reflect the data in the figure.
(5) Abstract and Introduction: Please describe the aim of this study, clearly.
(6) How does the loss of XBP1s influence on polyIC-stimulated cDC2 (including cytokine production)? The role of XBP1 is predominant in cDC1?
Author Response
This study addressed the impact of IRE1 and XBP1 on poly IC (a virus agonist)-induced cDC1 activation. The rational of this study is understandable. cDC1s have gained attention due to their ability to cross-present viral and tumor antigens to CD8 T cells. Elucidation of the activation mechanism of cDC1 is critical to manipulate the cells for protection from virus infection and establishment of tumor immunity. However, this reviewer fells that the data does not support conclusion in this study.
Major points
(1) Line 301 and 302: The conclusion should be rephrased. IRE1 deficiency does not have impact on expression levels of co-stimulatory molecules and cytokine production. It does mean that this molecule is not an emerging regulator of cDC1 activation with viral agonists.
R: We appreciate the reviewer input and have expanded the discussion section to provide plausible explanations for our findings:
On the other hand, Poly(I:C) increases proinflammatory cytokine production in cultured cDC1s and these factors show dependency on XBP1s expression, as induction of IL-12, TNF and IL-6 and type I interferons are decreased in Poly(I:C) stimulated XBP1s deficient OP9-DL1 cDC1s. Remarkably, our data show that these cytokines are produced at normal levels in cells lacking IRE1 RNase, which also lack XBP1s transcriptional activity. These data are novel, as it differs from previous observations in macrophages, which are known to depend on XBP1s transcriptional binding to the promoter regions IL-6/TNF of for optimal cytokine production25,44. In contrast, the reduced cytokine production noticed in XBP1s deficient OP9-DL1 cDC1s may be explained alternative mechanisms which include IRE1 RNase hyperactivation, which we provide evidence that occurs in OP9-DL1cDC1 from XBP1ΔDC animals. How exactly IRE1 RNase activation (induced upon XBP1s genetic loss) but not IRE1 RNase deficiency leads to changes in cytokine production is an issue that remains to be determined, and we speculate that it could be mediated directly by known IRE1 RNA substrates, such as microRNAs 40,45, or indirectly, by compensatory cross-activation of other UPR branches such as the PERK axis (Tavernier et al 2017, Nat. Cel. Bio). Taken together, these results show that a fine regulation of the IRE1/XBP1s axis controls cDC1 activation with viral agonists.
We hope the revised text now answers the reviewer request.
(2) The expression levels of VenusFP were significantly different between cDC1 and cDC2, whereas the expression levels of co-stimulatory molecules were similar in the two DC subsets. Therefore, it is not easy to assume that spliced XBP1 plays a crucial role in the expression of co-stimulatory molecules in cDC1. Please comment about it.
R: We agree with the reviewer and have now refined the text, to highlight that the correlation of activation and further increase of IRE1 RNase activity seems to be a selective feature of cDC1s
To extend these findings and identify efficient IRE1 activators in differentiated cDC1s, we stimulated OP9-DL1 cDCs from ERAI mice with agonists of major families of PRRs: LPS (TLR4 agonist), Poly I:C (TLR3 agonist), Curdlan (Dectin-1 agonist), DMXAA (STING agonist), MDP (NOD2 agonist) and CpG (TLR9 agonist). Data in Fig. 2A shows that cDC1s display higher VenusFP fluorescence than cDC2 in both basal levels and upon activation with all the microbial stimuli tested. We also observe that LPS and agonists of viral nature (Poly I:C, DMXAA and CpG) further induce marked VenusFP expression in OP9-DL1 cDC1s, although DMXAA stimulation also led to increased cell death (data not shown). These agonists also induce expression of the costimulatory molecule CD86 (Fig. 2B). However, this observation is not recapitulated in OP9-DL1 cDC2s, as these cells only induce VenusFP in response to CpG but show CD86 upregulation in response to LPS, Poly I:C, DMXAA and CpG (Fig. 2B). Thus, we conclude that cDC1 and cDC2s differ in their capacity to induce IRE1 RNase upon activation. For subsequent studies, we focused our analysis in Poly (I:C) stimulation, as it selectively activates IRE1 in the cDC1 compartment and it does not affect cell viability (Fig. 2C).
We hope the revised text now answers the reviewer request.
(3) Please explain the novelty of XBP1-related data.
Previous work has shown that XBP1s control proinflammatory cytokine production in macrophages, through direct binding of the reporter regions of TNF and IL-6 promoters (Martinon et al 2010, Nat. Imm). Our data differs with this evidence, as we show opposite results in two independent systems that lack XBP1s transcriptional activity (XBP1ΔDC and IRE1truncDC cDC1s lack XBP1s, yet only XBP1ΔDC cells show decreased cytokine production). Thus, our data highlights differences in cytokine regulation in macrophages and cDC1s. To clarify the statement we included the following sentence in the discussion section:
On the other hand, Poly(I:C) increases proinflammatory cytokine production in cultured cDC1s and these factors show dependency on XBP1s expression, as induction of IL-12, TNF and IL-6 and type I interferons are decreased in Poly(I:C) stimulated XBP1s deficient OP9-DL1 cDC1s. Remarkably, our data show that these cytokines are produced at normal levels in cells lacking IRE1 RNase, which also lack XBP1s transcriptional activity. These data are novel, as it differs from previous observations in macrophages, which are known to depend on XBP1s transcriptional binding to the promoter regions IL-6/TNF of for optimal cytokine production25,44. In contrast, the reduced cytokine production noticed in XBP1s deficient OP9-DL1 cDC1s may be explained alternative mechanisms which include IRE1 RNase hyperactivation, which we provide evidence that occurs in OP9-DL1cDC1 from XBP1ΔDC animals
Others
(1) Title should be rephrased.
R: We have rephrased the title; we hope now the reviewer finds it suitable.
(2) Fig. 3C: Please include FACS histogram data of CD80, CD40, PD-L1.
R: We have included the data in Figures 3C and 5D, as suggested.
(3) Line 205: Is it Fig.4?
R: It was figure 3. We thank the reviewer for noticing this error.
(4) The legend title of Fig. 4: Please check whether this title reflect the data in the figure.
We have adapted the title to:
Figure 4. XBP1s deficiency results in IRE1 RNase hyperactivation and RIDD in OP9-DL1 cDC1s.
(5) Abstract and Introduction: Please describe the aim of this study, clearly.
Abstract now includes the aim of the work in an implicit manner: Thus, the aim of this work is to elucidate if IRE1 RNase activity can also be modeled in cDC1s differentiated in vitro and reveal the functional consequences of such activation in cells stimulated with viral components.
Introduction now also states the aim of the study: Thus, the aim of this work is to demonstrate if cDC1s differentiated in vitro can also be regulated by IRE1 RNase activity, as there is great interest in targeting the subset for biomedical applications (Johnson et al 2022, Expert Opin Biol Ther)
(5) How does the loss of XBP1s influence on polyIC-stimulated cDC2 (including cytokine production)? The role of XBP1 is predominant in cDC1?
Being the work mostly focused on cDC1, we did not analyze the influence of the loss of XBP1 on the expression of costimulatory molecules in cDC2, as we magnetically separated cDC1s to rule out the influence that cDC2 could have on the results. However, we did measure intracellular cytokines production by cDC2s deficient for IRE1 or XBP1 stimu;ated with Poly I:C. However, these cells did not did not produce IL-12p40, IL-6 or TNFa in response to Poly(I:C), which may be probably due to tthe lack of TLR3 in the cDC2 compartment (Desch et al 2011, J. Exp. Med). Thus, we decided to leave those results out of the paper and focus only on cDC1. As previously reported (Medel et al., 2019; Osorio et al., 2014; Poncet et al., 2021; Tavernier et al., 2017) and the results obtained in our work XBP1 seems to have a predominant role in cDC1, as it regulates both survival and functionality of these cells in basal condition and in activation or infection contexts, while cDC2s do not seem to be affected by the absence of IRE1 nor XBP1s. However, it has been shown in intratumor cDC2-like cells that XBP1 have a rather detrimental role as it prevents the efficient presentation of tumor antigens (Cubillos et al., 2015).
Round 2
Reviewer 3 Report
No further comments